# Polarization-Insensitive Beam Splitter with Variable Split Angles and Ratios Based on Phase Gradient Metasurfaces

**DOI:** 10.3390/nano12010113

**Published:** 2021-12-30

**Authors:** Quan He, Zhe Shen

**Affiliations:** School of Electronic and Optical Engineering, Nanjing University of Science and Technology, Nanjing 210094, China; hequan@njust.edu.cn

**Keywords:** beam splitter, Snell’s law, phase gradient, metasurface

## Abstract

The beam splitter is a common and critical element in optical systems. Traditional beam splitters composed of prisms or wave plates are difficult to be applied to miniaturized optical systems because they are bulky and heavy. The realization of the nanoscale beam splitter with a flexible function has attracted much attention from researchers. Here, we proposed a polarization-insensitive beam splitter with a variable split angle and ratio based on the phase gradient metasurface, which is composed of two types of nanorod arrays with opposite phase gradients. Different split angles are achieved by changing the magnitude of the phase gradient based on the principle of Snell’s law of refraction, and different split ratios are achieved by adding a phase buffer with different areas. In the designed four types of beam splitters for different functions, the split angle is variable in the range of 12–29°, and the split ratio is variable in the range of 0.1–1. The beam splitter has a high beam splitting efficiency above 0.3 at the wavelength of 480–600 nm and a weak polarization dependence. The proposed beam splitter has the advantages of a small size and easy integration, and it can be applied to various optical systems such as multiplexers and interferometers for integrated optical circuits.

## 1. Introduction

The beam splitter is a type of optical element capable of splitting a beam of light into two or more ones. It has crucial applications in optical systems such as interferometers, spectrometers, and optical communications due to its abilities to split the polarization, frequency, and energy of light [1,2,3,4,5]. The classic beam splitter can be obtained with two glued triangular prisms or gratings [6,7,8]. Other structures were also widely used to make different types of beam splitters, such as waveguides [9,10,11], photonic crystals [12,13,14], wave plates [15,16], and multilayer films [17]. However, the aforementioned traditional beam splitters based on flat glass plates or prisms are bulky and heavy, which limit their further applications in compact optical systems. Therefore, they cannot meet the growing need for integrated beam splitters with the development of micro photonic devices.

The advance in nanophotonics provides a promising solution, which makes it possible to replace bulky optical devices with ultra-thin planar elements [18]. The metasurface is a two-dimensional ultra-thin artificial structure composed of sub-wavelength unit cells [19,20,21,22]. Due to the unique interaction between its wavelength-scale unit structure and light, metasurface can flexibly control the phase, polarization, and amplitude of incident light [23] and can even achieve functions that are not available from natural materials. The development of metasurface has made rich achievements, such as in planar lens [24,25], polarization devices [26,27], holograms [28,29], that have been gradually applied in various optical systems. In addition to the above mentioned basic optical elements, metasurfaces formed by the antenna array can be combined with the cavity to realize gain enhancement [30] and antenna bandwidth improvement [31]. The radiation pattern can be reconfigured by designing the structure of the antenna elements composing the metasurface [32]. It can be used to make artificial magnetic conductors with a 29% fractional bandwidth [33], transmit arrays for near-field transformation [34], and spatial filters with an ultra-wide stop-band [35]. The metasurface also can realize directivity improvement and phase correction of resonator antennas by algorithm optimization [36]. The composition of the metasurfaces are also diverse, including all-dielectric materials [37], printed materials [38], and all-metal surfaces [39]. The variety of preparations is conducive to its wide applications. The nanoscale beam splitter based on the metasurface can reduce the volume of optical devices and make optical systems more compact, which is of great significance in integrated optical circuits, such as on-chip optical interferometers and optical multiplexers.

There have been some reports of beam splitters based on metasurface with different methods [40,41,42,43,44], among which the more common is the design based on phase gradient metasurface. The phase gradient metasurface was first proposed by Capasso et al. [19], who used the arrangement of V-shaped antennas with different shapes to generate phase gradients to achieve beam deflection. It refers to the constant phase difference between adjacent unit cells on the metasurface, which can deflect the beam. At present, the phase gradient plays a crucial role in a variety of applications, typically in beam steering [45,46] and beam splitting. Furthermore, split angle and split ratio are two key indicators to evaluate the performance of a beam splitter. Flexible and variable split angles and split ratios make the beam splitter more suitable for optical systems. Ozer et al. [47] took advantage of the periodicity of the phase and designed a metasurface beam splitter using only the binary phase of 0 and π, which can split the light into two beams toward the left and right directions, respectively. However, the split angles are fixed and cannot be adjusted due to the use of only binary phases. Zhang et al. [48] used the upper and lower rows of nanorods with opposite phase gradients to achieve beam splitting. The split ratio can be changed by doping ions in one of the rows of nanorods. However, its essence is that the doped ions increase the loss of one of the split beams, which seriously affects the transmission efficiency of the beam splitter. Wei et al. [49] designed a metasurface with two parts of phase gradients which are arranged in opposite directions and have different magnitudes so that the two split angles could be different. Adjustable split ratio was achieved by moving the metasurface to change the proportion of incident light irradiated in the two opposite phase gradient areas, while this mobile working mode reduces its applicability in the actual fixed integrated optical circuit. In addition, Tian et al. [50] designed the phase gradients independently in the *x* and *y* directions to split the light into two mutually perpendicular planes. Additionally, the split ratio can be adjusted by controlling the ellipticity of the incident light. However, this beam splitter is polarization sensitive, which limits its general application. Therefore, it is still necessary to study polarization-insensitive beam splitters with a variable split angle and ratio.

In this work, we proposed a beam splitter based on a polarization-insensitive phase gradient metasurface, which is composed of super unit cells including two rows of nanorod arrays with opposite phase gradients. It can split the incident light into two beams with variable split angle and ratio. We first designed two rows of phase gradients with different magnitudes and opposite directions to achieve different split angles. Then, a phase buffer was introduced in one of the rows to make the energies of the two split beams different. Additionally, the area of the buffer zone was changed to make the split ratio variable. According to whether the split angles and split energies are the same or not, we designed four different types of beam splitters, respectively, to verify our design method. Finally, we used *x* and *y* polarized incident lights to study the polarization dependence of the proposed beam splitter and used incident light of different wavelengths to explore its broadband characteristic. Our works prospectively have the following novelties: (1) a new approach is proposed for the design of a beam splitter with flexible tuning capability based on phase gradient metasurface; (2) a concept of a phase buffer is proposed for the first time and used to adjust the split ratio; and (3) the proposed beam splitter can adjust the split angle and ratio simultaneously to make them variable with high transmission efficiency.

## 2. Theory and Designs of Metasurface

### 2.1. Generalized Snell’s Law

The phase gradient metasurface for beam splitting can be designed according to the generalized Snell’s law [19]. The relationship between the refracted angle of the transmitted beam and the phase gradient can be written as:(1)$$n_{t}\sin\theta -n_{i}\sin\theta_{0}=\frac{\lambda }{2\pi }\frac{d\varphi }{dx},$$
where $n_{t}$ and $n_{i}$ are the refractive index of the transmission medium and the incident medium, θ and $\theta_{0}$ are the refracted angle and incident angle, λ is the wavelength of the incident beam, and dφ/dx is the phase gradient attached to the metasurface. It can be seen that the refracted angle can be controlled by the additional phase gradient of the metasurface. In fact, the unit cells of the metasurface are not continuous, causing the phase gradient to be discrete. We can replace the continuous phase gradient dφ/dx with an *N*-step phase gradient 2π/Np. Additionally, the light is incident perpendicularly on the metasurface, $\theta_{0}=0$, so Equation (1) can be simplified as:(2)$$n_{t}\sin\theta =\frac{\lambda }{Np},$$
where *N* is the steps of phase from 0 to 2π, and *p* is the interval of two adjacent steps. According to Equation (2), we can see that when the step interval *p* is fixed, the refracted angle *θ* of the transmitted beam depends on the number of phase steps *N*.

### 2.2. Design of Metasurface

The schematic of beam splitting by a super-cell composing metasurface-based beam splitter is shown in Figure 1a. The super cell is composed of a nanoscale high refractive index dielectric cylindrical array and a SiO_2_ substrate. The structure of a single nanorod is shown in Figure 1b. Here, TiO_2_ material was selected to make nanorods because of its high real part of the refractive index and low loss in the visible waveband. The refractive indices of TiO_2_ and SiO_2_ are $n_{TiO_{2}}=2.43$ and $n_{SiO_{2}}=1.46$, respectively. In order to enable the *N*-step phase gradient generated by the nanorods to cover the phase range of 0 to 2π, we designed the size parameters of the unit cell. According to ref. [51], the height *h* of the nanorod and the period *p* of the unit cell were determined. At the working wavelength of 532 nm, the height *h* needs to meet $h>\lambda /(n_{TiO_{2}}-1)$ for making the TiO_2_ nanorod have the ability to produce a maximum of a 2π phase delay to the incident light. In order to ensure high diffraction efficiency, there should be only 0-order diffraction when the light is normal incident [52]. Therefore, the period *p* of the unit cell should be less than the equivalent wavelength of light in the substrate ($\lambda /n_{SiO_{2}}$) and greater than the diffraction condition ($\lambda /2n_{SiO_{2}}$). Finally, we chose the constants of *h* = 600 nm and *p* = 250 nm for high transmission efficiency and 0–2π phase coverage.

According to the cylindrical waveguide theory [53], the phase transmission of a nanorod is related to its effective refractive index, which varies with the radius of the cylinder, so the phase change from 0 to 2π can be achieved by changing the radius *r* of the nanorod. Figure 1c is the normalized magnetic energy density distribution of two nanorods with a distance of 250 nm and a radius of 100 nm in a nanorod array. It is obtained by the simulation of commercial software Lumerical-FDTD solutions (2018a, Lumerical Solutions, Vancouver, Canada) under 532 nm of light excitation. In this simulation, periodic boundary conditions were set in the *x* and *y* directions of the structure composed of two nanorods, and the perfectly matched layer (PML) boundary conditions were set in the *z* direction. Then, we placed a monitor on the *xoz* plane to observe the field distribution of the propagation section. It can be seen that when light passes through the nano-array, it is mainly limited to the inside of the nanorods due to the waveguide effect. Additionally, there is almost no coupling between adjacent nanorods, indicating that each nanorod can independently regulate the phase. Figure 1d shows the progress of the phase change in propagation for a plane wave with *λ* = 532 nm incidence under *r* = 100 nm. In order to select a suitable range of radius *r* to achieve 0–2π phase coverage and high transmission efficiency, we simulated nanorods with different radii. The relationships between the phase delay, the transmission of the unit cell, and the radius of the nanorod are shown in Figure 1e. Therefore, the phase coverage of 0–2π can be achieved by changing the radius of the nanorods from 40–110 nm with high transmissions.

As shown in Figure 1a, a super cell periodically arranged in the *x* and *y* directions as a metasurface is composed of upper and lower rows of nanorods, which produce opposite phase gradients in the *x* direction. Our designs can be divided into the following two types in terms of beam split energy. The first type is equal beam split energy, and *M* = 0. The number of nanorods with different radii in upper and lower rows are *N_l_* and *N_r_*, respectively, which form 0–2π periodic linear phase gradients in order to split the incident beam. The alignment directions of the upper and lower rows are opposite, which means that the phase gradients are opposite, resulting in the opposite beam split directions. The different split angles can be realized by making the number of *N_l_* and *N_r_* different. The second type is unequal split energy, and *M* ≠ 0. The upper row arrangement remains unchanged. In the lower row, besides *N_r_* nanorods used to form a phase gradient, there are also *M* nanorods with the same radii for 2π phase delay composing a phase buffer. The phase buffer has no phase gradient and will not refract the incident beam, so that it can weaken the energy of the rightward refracted beam. So, the energies of leftward and rightward split beams are different, and the number of *M* can be changed to achieve different split ratios. Our phase gradient design is mainly concentrated in the *x* direction, but the *y* direction will inevitably produce a phase gradient due to the arrangement of nanorods with different radii. In order to reduce the influence of the undesired phase gradient in the *y* direction on the split angle in *x* direction, we have repeatedly arranged 4 times in the y direction for each row of the designed phase gradient. At this time, the number of nanorods of a super cell in the x direction is the least common multiple of *N_l_* and *N_r_ + M*, and the number of nanorods in the y direction is 8. For the super cell, periodic boundary conditions are also used in the *x* and *y* directions, and PML conditions are used in the *z* direction. Then a monitor is placed on the *xoy* plane at a distance below the nanorod array to observe the transmitted field. It is worth noting that although the super cell splits the incident beam into two areas, its structure is wavelength scale, so the two refracted beams divided by the entire metasurface composed of periodic arranged super cells can still be regarded as plane waves.

## 3. Results and Discussions

### 3.1. Beam Splitting with Same Split Angle and Energy

Firstly, beam splitters with the same split angle and energy were designed. There is no phase buffer in a super cell, that is *M* = 0 in Figure 1a. Additionally, *N_l_* = *N_r_*, while the alignment directions of the nanorods are opposite. Therefore, the same number of nanorods in two rows guarantees that the magnitudes of the phase gradients are equal, so the magnitudes of the split angles are equal. In addition, the areas of the two regions with opposite phase gradients are the same, so the incident beam can be split into two beams with equal energy. We designed three groups of structures for *N_l_* = *N_r_* = 3, *N_l_* = *N_r_* = 4, and *N_l_* = *N_r_* = 5, respectively. In the three structures, the nanorods are periodically arranged along *x* direction, and the numbers of them covering a 0–2π phase period between upper and lower rows are the same. The phase increment of adjacent nanorods Δ*φ* along the +*x* direction is −2π/*N_l_* in the upper row, and it is 2π/*N_r_* in the lower row. The far-field projections of their normalized transmission fields are shown in Figure 2a–c, respectively, where the abscissa $U_{x}=\sin\theta$ is related to the split angle in the *xoz* plane. We can see that the transmitted light is split into two opposite directions close to the *xoz* plane. The split plane we considered is the *xoz* plane, and the theoretical value of *U_y_* is 0. However, it can be seen that there is also a weak split phenomenon in the *y* direction due to the inevitable phase gradient in the *y* direction, which is why the two split transmitted beams are not completely in the *xoz* plane. We regarded the transmitted field as the result of diffraction and calculated the normalized field intensities corresponding to the main split angles through the grating diffraction order method, as shown in Figure 2d–f. The split angles obtained by the simulation in Figure 2d–f are consistent with the angles $\theta_{N=3}=29.1{}^{\circ}$, $\theta_{N=4}=21.4{}^{\circ}$, and $\theta_{N=5}=16.9{}^{\circ}$, which were calculated according to Equation (2). We can intuitively conclude that the two split beams in the *xoz* plane have the equal magnitudes of angles and equal intensities in two opposite directions. It is worth noting that, as shown in Figure 2d–f, the number of the diffraction order will increase with the increase of *P_x_*. The secondary diffraction orders will occupy a part of the energy of the transmitted light, resulting in an increase in stray light.

### 3.2. Beam Splitting with Different Split Angles and Same Split Energy

In a super cell composing the beam splitter with different split angles and same split energy, *N_l_* ≠ *N_r_* and *M* = 0. The period of the super cell in the *x* direction is the least common multiple of *N_l_* × *p* and *N_r_* × *p*. The inequality of *N_l_* and *N_r_* makes the magnitude of phase gradients in the upper and lower areas in Figure 1a unequal, but the directions of phase gradients are still opposite, so that the magnitude of refracted angles to left and right are not equal. The absence of a phase buffer ensures that the energies of the refracted beams to left and right are the same. When *N_r_* = 3, we designed three groups of structures for *N_l_* = 4, 5, and 6, respectively. At this time, the arrangement of the nanorods is similar to that in Section 3.1, but the difference is that the numbers of nanorods covering a 0–2π phase period between upper and lower rows are different. In three structures, the phase increments of adjacent nanorods Δ*φ* along the +*x* direction in the upper row are −π/2, −2π/5, and −π/3 respectively, and it is fixed at 2π/3 in the lower row. The corresponding far-field projection intensity distributions are shown in Figure 3a–c. The transmitted light is well split into two beams. The normalized intensities corresponding to the refracted angles in the three sets of results are shown in the lower row of Figure 3. As in Figure 3d–f, the angles *θ_r_* of the beam refracted to the right are 29.1° due to *N_r_* is 3. Additionally, *θ_l_* are −21.4°, −16.9°, and −14.1° corresponding to *N_l_* equal to 4, 5, and 6. The magnitude relationships between these different *N_l_* or *N_r_* and *θ_l_* or *θ_r_* all satisfy Equation (2). As *N_l_* increases, *θ_l_* will decrease because of the decreasing phase gradient. In the three results of beam splitting, the split ratios $sr=E_{r}^{2}/E_{l}^{2}$ at the main angles are 0.95, 0.89, and 0.91, respectively, which are close to one. That they are not exactly equal to one can be explained as follows. In this type of anomalous refraction based on phase gradient metasurface, the smaller the phase gradient, the higher energy of the refracted light. This may be due to the large number of discrete phase sampling points in the 0–2π period, which makes the step phase gradient 2π/Np more continuous, so it is more suitable for the generalized Snell’s law of refraction. In other words, the realization of a large, refracted angle comes at the cost of split energy. In Figure 3d–f, $E_{r}^{2}<E_{l}^{2}$ due to *N_r_ < N_l_* and $E_{r}^{2}$ stays almost unchanged due to the same *N_r_*. Theoretically, the energy difference between the left and right split beams is more obvious as *N_l_* increases, so *sr* is smaller. Thus, *sr* in Figure 3d is the largest. However, *sr* in Figure 3e is slightly smaller than that in f because it is also affected by the number of diffraction orders of different super cells (the number of diffraction orders in Figure 3e is more than that in Figure 3f). In general, we have achieved beam splitting with different split angles and nearly the same split energy.

### 3.3. Beam Splitting with Same Split Angle and Different Split Energies

We then designed beam splitters with the same split angle and different split energies. For the design of different split energies, it is necessary to introduce a phase buffer in one of the beam splitting directions, that is, *M* ≠ 0. In order to have the same magnitude of split angle after the phase buffer is introduced, the magnitude of phase gradients in the two beam splitting directions should still be equal. As shown in Figure 1a, the *N_l_* columns of nanorods still cover the 0–2π phase range in order to refract to the left. Phase buffer composed of *M* columns of nanorods with a uniform 2π phase delay is added in the area for beam splitting to the right. Additionally, the magnitude of phase gradient caused by *N_r_* columns of nanorods for rightward refraction is equal to that by *N_l_* columns for leftward refraction. At this time, *N_l_ = N_r_ + M*, and the period of the super cell in the *x* direction is *N_l_* × *p*. In this design, not only the magnitude of phase gradients of the areas composed of the *N_l_* columns and the *N_r_* columns are equal, but also the magnitude of average phase gradients of the areas composed of the *N_l_* columns and the *N_r_ + M* columns are equal. Here, we have designed four structures with *N_l_* = 7 and *M* as 1, 2, 3, and 4 respectively. In these structures, it can be regarded as that the nanorods all having a 2π phase delay replace the *M* adjacent nanorods that form the phase gradient in the lower row of the designs in Section 3.1 with *N_l_* = *N_r_* = 7. In the upper row, the phase increment of adjacent nanorods Δ*φ* along the +*x* direction is fixed at −2π/7. In the lower row, there are *N_r_* nanorods with adjacent 2π/7 phase increments and *M* nanorods all with a 2π phase delay in an arranged cycle. Their far-field projection intensity distributions are shown in Figure 4a–d. The four sets of results all show the good ability of leftward beam splitting, while the ability of rightward beam splitting gradually weakens with the increase of *M*, and the stray light gradually increases significantly. The normalized intensity distributions of the main refracted angles are shown in Figure 4e–h. In our design, (*θ_l_*, *θ_r_*) is fixed at (−12.0°, 12.0°), which is consistent with the theoretical calculation of Equation (2) when *N* = 7. In Figure 4e, *M* = 1, which has a small effect on the difference of split energy, and *sr* = 0.95 indicating that it is still close to the beam splitting with the same split energy. However, as the number of columns of the phase buffer increases, on the one hand, the continuity of the overall phase gradient is weakened; on the other hand, its feature of no phase gradient prevents refraction at a non-zero degree. Therefore, the effect of rightward beam splitting is weakened by the presence of the phase buffer, which is shown in Figure 4f–h as the corresponding light intensity gradually decreases at the designed split angle *θ_r_*. The reduced part of the light is diffracted to other diffraction orders, resulting in an increase of stray light. The increase in light intensity under *θ_l_* in Figure 4f–h compared to e may be due to the superposition of the stray light. In addition, as shown in Figure 4e–h, the intensity corresponding to 0° gradually increases. This is because the vertically transmitted light intensity increases with the increase of the phase buffer area. Therefore, we have realized beam splitters with the same split angle and different split energies, and the split ratio is variable within 0.1 and 1.

### 3.4. Beam Splitting with Different Split Angles and Energies

Finally, we designed beam splitters with different split angles and different split energies. The difference from the design in Section 3.3 is that the magnitudes of phase gradients of the upper and lower areas are also different after the phase buffer is introduced in one of the areas. In the design of Figure 1a, *N_l_* = *N_r_*, and the phase buffer composed by *M* columns of nanorods is added in the area for rightward beam splitting. The period of the super cell in the *x* direction is the least common multiple of *N_l_* × *p* and (*N_r_ + M*) × *p*. In this case, even if *N_l_* and *N_r_* are equal, while the magnitudes of average phase gradients of leftward and rightward beam splitting areas are not equal due to the introduction of the phase buffer, resulting in different magnitudes of split angles. Additionally, the presence of the phase buffer will weaken the light intensity of the rightward refracted beam so that the split energies are different. We have designed four structures with *N_l_* = 3 and *M* as 1, 2, 3, and 4, respectively. Here, it can be seen as adding *M* nanorods all with a 2π phase delay to the low row of the design in Section 3.1 with *N_l_* = *N_r_* =3. In the upper row, the phase increment of adjacent nanorods Δ*φ* along the +*x* direction is fixed at −2π/3. In the lower row, there are three nanorods with adjacent 2π/3 phase increments and *M* nanorods all with 2π phase delays in an arranged cycle. The corresponding far-field projection intensity distributions are shown in Figure 5a–d, respectively. Similar to the results in Figure 4, a good leftward beam splitting is maintained when *M* is a different value, while the energy of the rightward beam splitting decreases as *M* increases. Further, the intensity distributions corresponding to the main angles are shown in Figure 5e–h. When *M* is from one to four, (*θ_l_*, *θ*_r_) are (−29.1°, 21.4°), (−29.1°, 16.9°), (−29.1°, 14.1°), (−29.1°, 12.0°), respectively. The leftward split angle *θ_l_* remains unchanged, while the rightward split angle *θ_r_* gradually decreases due to the decrease of the magnitude of the average phase gradient. Figure 5e–h, the light intensity $E_{l}^{2}$ at *θ_l_* remains almost unchanged, while the $E_{r}^{2}$ at *θ_r_* gradually decreases, so the split ratio *sr* also decreases. It can be noticed that $E_{l}^{2}$ in Figure 5f–h is significantly larger than that in Figure 5e, and it is due to the superposition of stray light, which is the same as that in Section 3.3. Therefore, beam splitters with different split angles and energies are realized, and the split ratio can be changed in the range of about 0.2–0.8 by controlling the number of *N* and *M*.

### 3.5. Broadband and Polarization Insensitive Characteristics

We continued to explore the performance of the proposed beam splitter in waveband and polarization dependence. The structure designed with *N_l_* = *N_r_* = 5 in Section 3.1 is used for wavelength scanning from 450 nm to 700 nm. Figure 6a is the relationship between the transmission and wavelength for the *x* polarized light incidence, where *T_l_* and *T_r_* are the intensities of the leftward and rightward split beams respectively, and *T_c_* is the intensity of the vertical transmitted beam. We can see that between 480–600 nm, the *T_l_* and *T_r_* are equal and higher than 0.3, while *T_c_* is lower than 0.1, which shows that most of the light is refracted in the two desired directions. So, the beam splitter we designed has a certain broadband characteristic. Figure 6b shows the split angles obtained by simulation (solid line) and theoretical calculation (dashed line) at the corresponding wavelength. The magnitude of split angles *θ_l_* and *θ_r_* are always equal and positively correlated with the wavelength, which is caused by the dispersion characteristic of the metasurface. The agreement between the simulation and theoretical results proves that the generalized Snell’s law can be used to explain the metasurface we designed. In Figure 6a, *T_l_* and *T_r_* begin to decrease, while *T_c_* begins to increase when the wavelength is greater than 600 nm. This is because longer wavelength corresponds to larger split angle, and the realization of large split angle will come at the cost of split energy as we discussed in Section 3.2. On the contrary, the vertical transmittance is high at longer wavelength. In order to analyze the polarization dependence of the proposed beam splitter, *T_l_*, *T_r_,* and *T_c_* in 450–700 nm for *y* polarized light incidence are obtained as shown in Figure 6c. We can see that the beam splitter can still maintain a beam splitting with the ratio of 1:1 across the entire waveband. The change trends of the transmittance of the split beams with wavelengths are almost the same as that of the *x* polarized incident in Figure 6a. To further measure the polarization dependence, we calculated the difference of *T_l_* or *T_r_* between the *x* and *y* polarized lights incidences, as shown in Figure 6d. The magnitude of the difference is in the range of 0.1, indicating that the proposed beam splitter is weakly dependent on polarization. It is worth noting that although a single nanorod has a high rotational symmetry, the arrangement of the metasurface is still anisotropic in the *x* and *y* directions, causing the above-mentioned slight difference. In addition, the *T_l_* and *T_r_* in Figure 6c are generally slightly smaller than those in Figure 6a, which may be due to the fact that the incidence of *y* polarized light will enhance the diffraction in the *y* direction, while the light intensity we calculated is the region close to the *x* axis, and the light intensity far from the *x* axis will be ignored. However, diffraction behavior weakens at longer wavelength, so the difference between *T_x_* and *T_y_* in Figure 6d decreases between 600–700 nm.

## 4. Conclusions

In conclusion, we proposed a high efficiency polarization-insensitive beam splitter based on the phase gradient metasurface with a variable split angle and ratio simultaneously. The variable split angle within 29° is realized based on the principle of Snell’s law of refraction by controlling the number of nanorods covering the phase range of 0–2π. Additionally, the variable split ratio can be changed between 0.1 and 1 by adjusting the area of the phase buffer. The four types of designed beam splitters all show good beam splitting at a wavelength of 532 nm. The high agreement between the split angles obtained, respectively, by simulation and theoretical calculation shows the feasibility of our design method. In addition, the high split efficiency of the proposed beam splitter between 480–600 nm proves that it has a broadband function. Additionally, the consistency of the results under the incidence of *x* and *y* polarized lights indicates that the beam splitter is polarization-insensitive. This type of metasurface-based beam splitter is expected to be used widely in integrated optical systems, such as interferometers and multiplexers.

## Figures and Tables

**Figure 1 nanomaterials-12-00113-f001:**
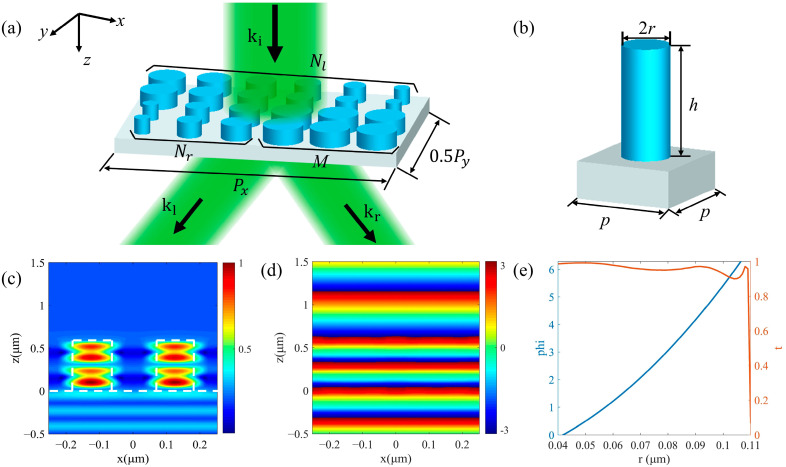
(**a**) Schematic of beam splitting by a super cell of beam splitter based on metasurface composed of nanorods. The beam is incident from the top through the metasurface and split into two beams in the *x* direction. (**b**) The 3D model of a single nanorod unit cell with radius *r*, height *h*, and period *p*. (**c**) The magnetic energy density distribution and (**d**) the progress of the phase change in the propagation of two nanorods with *r* = 100 nm, *p* = 250 nm by FDTD. The white dashed lines are the outline of the nanorods and the substrate. (**e**) The relationships between phase delay, transmission of the unit cell, and the radius of the nanorod.

**Figure 2 nanomaterials-12-00113-f002:**
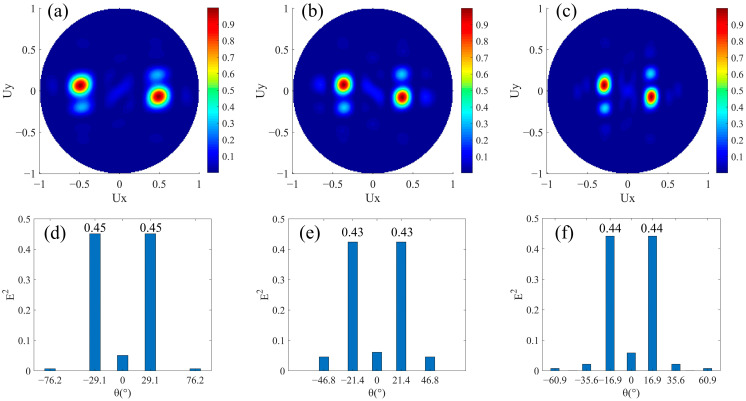
Simulation results of beam splitters with same split angle and energy. *N_l_* = *N_r_*, *M* = 0. The normalized far-field projection intensity distributions of designs (**a**) *N_l_* = *N_r_* = 3, (**b**) *N_l_* = *N_r_* = 4, and (**c**) *N_l_* = *N_r_* = 5, respectively. (**d**–**f**) are the normalized field intensity under the main split angle corresponding to the results of (**a**–**c**), respectively.

**Figure 3 nanomaterials-12-00113-f003:**
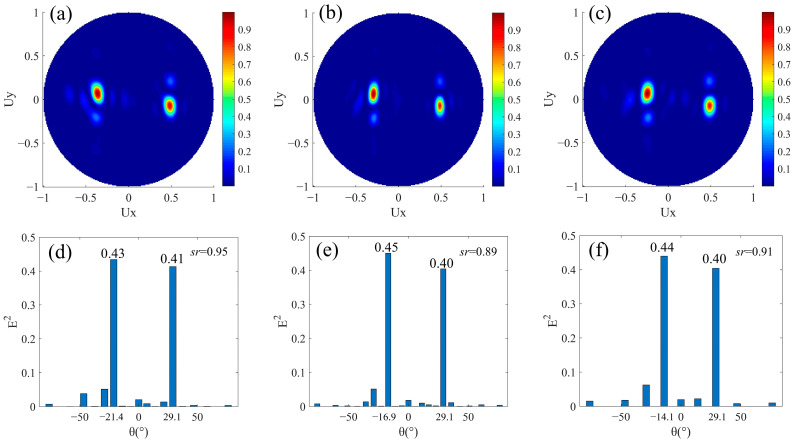
Simulation results of beam splitters with different split angles and same split energy. *N_l_* ≠ *N_r_*, *M* = 0. The normalized far-field projection intensity distributions of designs (**a**) *N_l_* = 4, *N_r_* = 3, (**b**) *N_l_* = 5, *N_r_* = 3, and (**c**) *N_l_* = 6, *N_r_* = 3, respectively. (**d**–**f**) are the normalized field intensity under the main split angle corresponding to the results of (**a**–**c**), respectively.

**Figure 4 nanomaterials-12-00113-f004:**
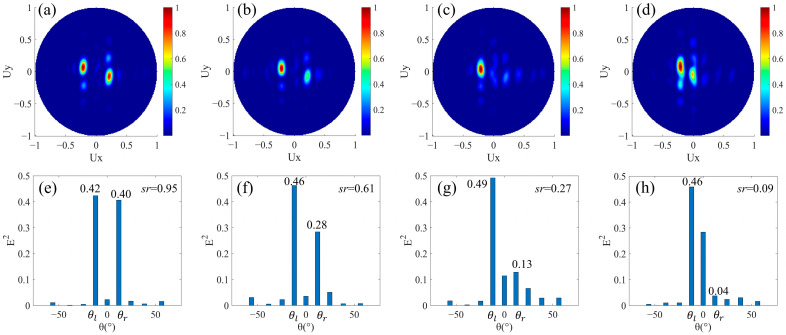
Simulation results of beam splitters with same split angle and different split energies. *N_l_* = *N_r_* + *M*, *M* ≠ 0. The normalized far-field projection intensity distributions of designs (**a**) *N_l_* = 7, *M* = 1, (**b**) *N_l_* = 7, *M* = 2, (**c**) *N_l_* = 7, *M* = 3, and (**d**) *N_l_* = 7, *M* = 4, respectively. (**e**–**h**) are the normalized field intensity under the main split angle corresponding to the results of (**a**–**d**) respectively.

**Figure 5 nanomaterials-12-00113-f005:**
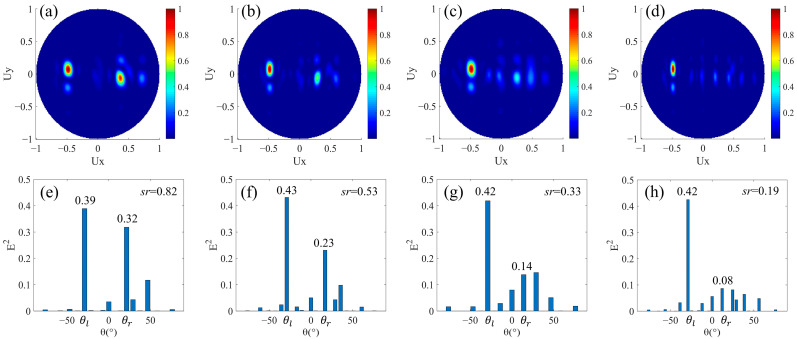
Simulation results of beam splitters with different split angles and energies. *N_l_* = *N_r_*, *M* ≠ 0. The normalized far-field projection intensity distributions of designs (**a**) *N_l_* = *N_r_* = 3, *M* = 1, (**b**) *N_l_* = *N_r_* = 3, *M* = 2, (**c**) *N_l_* = *N_r_* = 3, *M* = 3, and (**d**) *N_l_* = *N_r_* = 3, *M* = 4, respectively. (**e**–**h**) are the normalized field intensity under the main split angle corresponding to the results of (**a**–**d**), respectively.

**Figure 6 nanomaterials-12-00113-f006:**
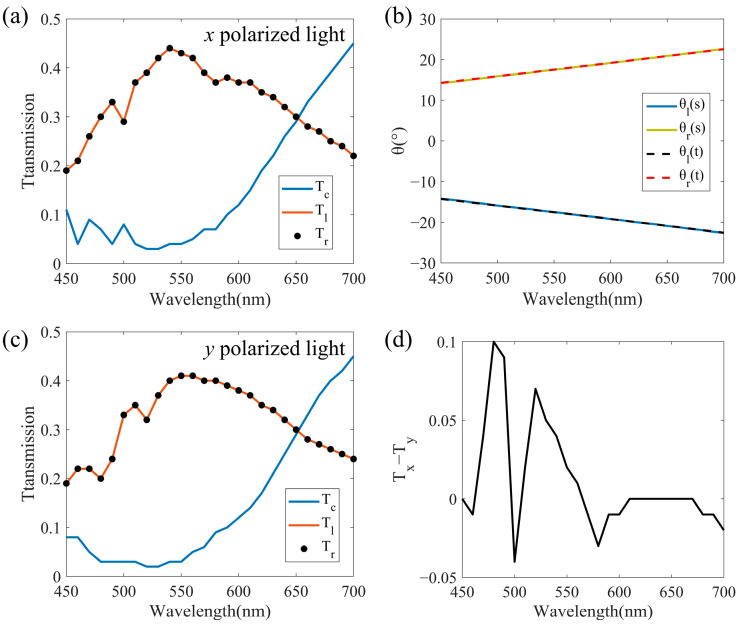
The results of verifying broadband characteristic and polarization insensitivity by the design of *N_l_* = *N_r_* = 5 in Section 3.1. (**a**) Transmittance spectrum of split leftward beam *T_l_*, split rightward beam *T_r_*, and vertical transmitted beam *T_c_* for *x* polarized light incidence. (**b**) The split angles obtained by simulation (solid line) and theoretical calculation (dashed line) at the corresponding wavelength. (**c**) Transmittance spectrum of *T_l_*, *T_r_*, and *T_c_* for *y* polarized light incidence. (**d**) The difference of *T_l_* or *T_r_* between the *x* and *y* polarized lights incidences.

## Data Availability

Data are contained within the article.

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
