# Peer review of "Polarization-Insensitive Beam Splitter with Variable Split Angles and Ratios Based on Phase Gradient Metasurfaces"

_nanomaterials, 2021, doi:10.3390/nano12010113_

Round 1
Reviewer 1 Report
The authors present their modelling and designs for metasurfaces to be used as beam splitters. They provide a background discussion on earlier beam splitter designs using metasurfaces. They then explain their design rationale, and the basis for the variable angle of the beam splitter and the variable power ratio. The wavelength response of the designs is approximately limited to the visible range.
I recognise that the authors have tried to explain their design process carefully. However, some elements of the explanation are not clear, approximately lines 123 to 185 need some work so that each element of the design is described accurately and clearly. Since the designs of the metasurfaces are the actual results of this paper, the authors need to provide these in detail. (I suggest providing full details for each of the devices illustrated in the figures).
Reviewer 2 Report
The authors reported the manuscript entitled " Polarization-insensitive beam splitter with variable split angles and ratios based on phase gradient metasurfaces" very well. Paper may be accepted after minor revision:
- English language must be checked thoroughly.
- The novelty of research work should be included in the manuscript.
- The conclusion must be shortened.
Reviewer 3 Report
A polarization-insensitive phase gradient metasurface with opposite phase gradients super cells is presented in this work. The metasurface is capable of splitting the incident light into two beams with different split angle and ratio. The paper is well-structured and the results look convincing. However, there are some aspects that need improvements and some ambiguities that need to be rectified. Here are the detailed comments:
Please add cross-polarization results.
Please explain the boundary condition used in Fig.1,2.
Please note Nanomaterials does not only focus on Electromagnetics and it has a broad readership, hence it is necessary to give a short intro on the metasurfaces and then narrow the topic down to beamsplitters.
The introduction is a bit abrupt and does not adequately umbrella the state-of-the-art in relation to metasurfaces in both aspects of manufacturing and application. A short introduction about metasurface should be added where the applications of metasurfaces are briefly mentioned. It should be mentioned that metasurfaces are used for gain enhancement as explained in Directivity improvement of a Fabry-Perot cavity antenna by enhancing near field characteristic. Metasurfaces are also used for antenna bandwidth improvement, which is explained in Single-dielectric wideband partially reflecting surface with variable reflection components for realization of a compact high-gain resonant cavity antenna. Another application of metasurfaces is reconfiguring radiation patterns of antennas as explained in Single-layer polarization-insensitive frequency selective surface for beam reconfigurability of monopole antennas; Metasurfaces can be used as artificial magenetic conductors that was explained in Design of an artificial magnetic conductor surface using an evolutionary algorithm. Another application of metasurafce is transmit array as discussed in All-metal wideband metasurface for near-field transformation of medium-to-high gain electromagnetic sources.
Metasurfaces also are used as spatial filters, as demonstrated in Single-layer ultra-wide stop-band frequency selective surface using interconnected square rings. Another recent application of metasurfaces is phase correction as discussed in (Multiobjective particle swarm optimization to design a time-delay equalizer metasurface for an electromagnetic band-gap resonator antenna).
Please elaborate on the simulation procedure and give more details about the simulation setup.
Results and discussion: Please explain the mechanism you captured the electric field.
Is it possible to extract a model from the metasurface?
In terms of metasurface manufacturing and prototyping, it needs to be mentioned that metasurfcaes can be made of all-dielectric materials such as A high-gain wideband ebg resonator antenna for 60 GHz unlicenced frequency band. Metasurfaces can be made of printed (hybrid) materials as explained Gain enhancement of wideband circularly polarized UWB antenna using FSS. They can be made of all-metal surfaces Low-cost nonuniform metallic lattice for rectifying aperture near-field of electromagnetic bandgap resonator antennas. All THREE categories (i.e. all-dielectric, printed layers, and all-metal) should be included in the Introduction with some relevant references.
The theoretical aspect of the paper can be improved.
What about the measurement?
Please explain the phase gradients in a bit more detail and mention the other applications of it, such as beam steering. The following papers are recommended to be included in this regard:
“Beam-Scanning Antenna Based on Near-Electric Field Phase Transformation and Refraction of Electromagnetic Wave Through Dielectric Structures”, IEEE Access 8, 199242-199253
“Low-RCS beam-steering antenna based on reconfigurable phase gradient metasurface”. IEEE Antennas and Wireless Propagation Letters, 18(10), pp.2016-2020.
In summary:
The important application of metasurfaces (gain enhancement, bandwidth improvement, reconfiguring radiation patterns,…) should be mentioned with relevant refs.
A short explanation of the construction of metasurfaces (all-metal, all-dielectric and hybrid) needs to be added
More explanation on design as commented above.
Round 2
Reviewer 3 Report
There is no more concern.